# Priority healthcare needs amongst people experiencing homelessness in Dublin, Ireland: A qualitative evaluation of community expert experiences and opinions

Carolyn Ingram[1]*, Isobel MacNamara[1,2], Conor Buggy[1], Carla Perrotta[1]

1 Public Health, School of Public Health, Physiotherapy, and Sports Science, University College Dublin, Dublin, Ireland, 2 School of Medicine, University College Dublin, Dublin, Ireland

* Carolyn.ingram@ucd.ie

**Data Availability Statement:** All relevant data are within the paper and its Supporting Information files. Relevant data includes transcript excerpts

## Abstract

In light of evidence that housing-related disparities in mortality are worsening over time, this study aimed to explore the perspectives of experts working in homeless health and addiction services on priority healthcare needs amongst people experiencing homelessness in Dublin, Ireland, a city facing problematic increases in homelessness. As part of a larger qualitative study, a series of semi-structured interviews were carried out with 19 community experts followed by inductive thematic framework analysis to identify emergent themes and sub-themes relating to priority healthcare needs. At the societal level, community experts identified a need to promote a culture that values health equity. At the policy level, accelerating action in addressing health inequalities was recommended with an emphasis on strategic planning, Housing First, social support options, interagency collaboration, improved data linkage and sharing, and auditing. At the health services level, removing barriers to access will require the provision of more and safer mental health, addiction, women-centred, and general practice services; resolved care pathways in relation to crisis points and multi-morbidity; expanded trauma-informed education and training and hospital-led Inclusion Health programmes; and outreach programmes and peer support for chronic disease management. The voices of people experiencing homelessness, including representatives from specific homeless groups such as migrants, youth, and the elderly, must be thoroughly embedded into health and social service design and delivery to facilitate impactful change.

## Introduction

Rising homelessness levels across Europe have drawn increased attention to the health consequences of precarious housing. A health equity perspective–the belief that everyone has the right to a fair and just opportunity to attain their full potential for health–emphasizes that health disparities amongst people experiencing homelessness (PEH) are unnecessary and avoidable [1]. Rather than the result of biological differences, they result from social and economic

justifying the results and quotations reported in our manuscript. Researchers who meet access criteria may register to view anonymised transcripts via the Qualitative Data Repository (https://doi.org/10.5064/F6HFOEC5).

**Funding:** The authors received no specific funding for this work.

**Competing interests:** The authors have declared that no competing interests exist.

processes that create and recreate differences in access to housing and in turn health [1]. Low income and minority individuals and families are more vulnerable to the structural, institutional, relationship, and personal causes that drive homelessness [2]. Relationship breakdown as a result of domestic violence or abuse, or unemployment and financial stress, may leave someone without stable accommodation, as can the death of a family member [2]. Youth leaving foster care or those leaving prison or mental health facilities may have nowhere to go upon release or discharge [2]. People entering homelessness are more likely to have health issues, mental health issues, learning difficulties, problematic alcohol and drug use, or experiences of trauma in childhood [2]. In the Republic of Ireland, a country facing a severe housing crisis, structural causes exacerbate these housing inequalities by providing limited social housing and relying on the private rental market which drives rent spikes and insecure tenancies [2].

Once in homelessness, an individuals' health worsens and their life expectancy shortens [3]. PEH are at higher risk of premature mortality, disability, and chronic conditions and rely heavily on acute care [4, 5]. PEH in Ireland are at increased risk of substance misuse, self-harm, frequent emergency department (ED) presentation, and psychiatric admission. They are more likely to be hospitalised for venous thromboembolism (VTE), pneumonia/bronchitis, cellulitis, and seizure but less likely to complete their hospital admission [5]. Despite facing adverse health outcomes, PEH experience reduced access to care due to a variety of external barriers including physical proximity to services; administrative barriers such as queues, appointment systems, and complicated application processes for care; and stigma and judgement from healthcare providers [6]. Once able to access services, PEH still may not use them due to competing priorities (e.g., finding food or a place to sleep), presumption of poor treatment, or negative emotions like fear, embarrassment, hopelessness, and poor self-esteem [6].

Compounding housing and health inequalities stress the extent to which the individuals most likely to enter homelessness are the least equipped to combat its health consequences. In this context, achieving health equity will require not only addressing the downstream drivers of homelessness, but also improving equitable access to healthcare for PEH by strengthening health planning and the allocation of resources. This, first, requires understanding priority health concerns amongst PEH within a shared system [7].

In autumn 2022, multiple factors highlighted a need to prioritise healthcare needs amongst PEH in Dublin. First, Ireland is unique in Europe in the severity of recent spikes in homelessness with three quarters of homeless individuals residing in the capital city [8]. Second, evidence shows that disparities in mortality between homeless and housed populations are worsening over time. In Dublin, standardised mortality ratios (SMRs) are more than 10 times higher in PEH compared with the general population [3]. Finally, the Covid-19 pandemic acted as a catalyst for change in the delivery of harm reduction measures to homeless individuals who use drugs. Two of those changes (the removal of barriers to rapid access to methadone and the expanded distribution of Naloxone) demonstrated the capacity for policymakers to facilitate change in response to a strong public health argument [9]. Researchers saw the potential for positive change and sought to provide further evidence on how and where to allocate health and social service resources effectively.

Qualitative methods can provide timely insights into pressing health and social care needs as they can be conducted within a short time period and with involvement of community experts and members from marginalised groups [7, 10]. Integrating qualitative techniques into local health needs assessments enables policymakers and healthcare providers to work out which interventions are needed most in their area; to plan strategically what services are needed in the future; to work out where resources should be focused based on evidence; and to address health inequalities [11]. Guidelines specific to homeless health needs assessments recommend asking both people who are homeless as well as professionals who deliver services to

people who are homeless to identify gaps in healthcare provision, barriers, and solutions [11]. Research teams from Calgary and San Francisco have demonstrated the efficiency and effectiveness of this approach, employing semi-structured interviews with staff in homeless health and service organisations to help identify priority health and healthcare needs amongst PEH across each city [12, 13]. The aim of this facet of a larger qualitative study [14] was to explore the perspectives of experts working in homeless health and addiction services in Dublin on priority healthcare needs amongst PEH using qualitative semi-structured interviews. To ensure the integration of community voices [15], health concerns of homeless service users in Dublin were explored through ethnographic field work and will be reported as a separate paper within the larger study.

## Materials and methods

### Study design and setting

The study's target population comprised health and social care professionals, government health agents, and addiction services employees who have worked for at least five years with PEH in Dublin City. Under Ireland's Housing Act 1988, homelessness is defined as (1) having no accommodation available that can reasonably be stayed in or living in a hospital, county home, night shelter or other such institution, and (2) being unable to provide accommodation from one's own resources [16]. In Dublin City (population 588,233), approximately 6,305 adults and 2,964 children are currently residing in emergency accommodation [17]. An estimated 80 individuals are sleeping rough [18], and up to 6% of the city's residents are experiencing hidden homelessness (i.e., sleeping cars, in squats, on the floors or sofas of family and friends, or in unsafe accommodation [19]. Drug and alcohol use is the primary driver of homelessness in Ireland and remains a leading cause of death within the homeless population [20]. Most of the country's homeless and addiction services are concentrated in Dublin inner city comprising the city centre and its immediately surrounding neighbourhoods within two canals, reflecting the gravity of need within this area.

Qualitative approaches are recommended for exploring health needs specific to underrepresented, marginalised segments of the community [10]. Semi-structured interviews with community experts (CE) allow for in-depth understanding of community history, social activities, local resources and gaps in services [21]. Thus, we conducted qualitative, semi-structured interviews with CEs. During the study period, the lead researcher simultaneously conducted 60 hours of participant observation at a drop-in primary care and addiction service during which they routinely spoke with homeless service users (Ethics Reference: LS-22-41-Ingram-Perrotta). These interactions informed the selection of CEs, design of interview questions, and allowed us to contextualise and better reflect upon interview findings.

### Sample and recruitment

We used a purposive, criterion-i/snowball sampling strategy to identify professionals working in homeless health and/or addiction services in Dublin, stratified by occupation type. Criterion-i sampling refers to the identification and selection of cases that meet a predetermined criteria of importance [22] (in this case, providing services to PEH in Dublin). We identified potential CEs through (1) an internet search of homeless health and addiction services in Dublin, and (2) based on recommendations made by community partners working in the primary care and addiction service clinic where the lead researcher conducted her fieldwork. We also invited interviewed CEs to recommend colleagues they felt would have relevant perspectives on community health needs. In total, we contacted 30 potential CEs via phone or email between 01 August 2022 and 31 March 2023, of which 19 agreed to participate in an interview.

## Data collection

We conducted semi-structured interviews between September 2022 and March 2023 utilising ZOOM™, the phone, or in person according to participant preference. The interviewer, CI, has formal qualitative research training. We presented CEs with an information sheet and sought audio recorded, informed oral consent –considered appropriate for remote research conducted with non-vulnerable participants [23] - in the full knowledge that interviews would be audio recorded, transcribed, and anonymised as approved by our institutional Human Research Ethics Committee (LS-E-125-Ingram-Perrotta-Exemption). We based interview questions on World Health Organization Community Health Needs Assessment guidelines [24] and refined questions over time based on perspectives gained from other interviews and the lead researcher's field work with homeless service users (Table 1). For example, after hearing PEH mention experiencing traumatic events after which they had nowhere to go/no one to turn to, we added, *"Have you spotted any gaps in services?"* Or, after multiple professionals in addiction services mentioned an overreliance on methadone, we added a question about methadone prescribing in subsequent interviews with primary care providers.

## Data analysis

We coded and analysed data using an inductive thematic framework method according to the following recommended stages of trustworthy, thematic analysis [25, 26]:

- **Transcription**: CI transcribed audio recordings verbatim.

- **Familiarisation**: Two researchers (CI, IM) familiarised themselves with the data by re-reading the transcripts. Each researcher recorded analytical notes, thoughts, and impressions in the transcript margins.

- **Identification of a Thematic Framework**: The same researchers independently coded three transcripts before meeting to discuss key themes and constructing an initial coding framework. Using this framework, CI and IM coded three more transcripts each before further discussing, revising and refining the work. This process was repeated six times until no new themes were generated.

- **Applying the thematic framework**: Using NVivo™ software (Version 11), CI systematically applied the working thematic framework to all transcripts before organising codes into categories reflecting priority healthcare needs.

**Table 1. Community healthcare needs topics included in the semi-structured interview guide.**

| Topic | Guiding Questions |
|---|---|
| Work Focus & Experience | • What does your work currently focus on and with what populations? |
| Priority Areas of Action | • What changes have you seen take place in the community and the services you provide?<br>• What programmes do you see working well? Or less well?<br>• What changes would you make if you could? |
| Healthcare Access & Use | • Have you spotted any gaps in services?<br>• Are most homeless people using provided services? |
| Understanding the Community | • Which health issues are of most concern to the communities you work with?<br>• What are the strengths and assets in these communities? |
| Gaps in Research | • Where do we need more evidence?<br>• What would you focus your next research project on? |

- **Critical friend approach**: CI met with CB and CP to critically review and discuss codes, themes, and sub-themes using a critical friends approach. 'Critical friends' serve as a theoretical sounding board to encourage reflection upon alternative pathways and explanations [27]. These discussions led to the merging of existing codes.

- **Charting and interpreting the data**: CI used a matrix to summarise data for each CE, code, and theme. Connections within and between codes and cases were made in order to explain priority healthcare needs, their causes, and implications.

- **Member checking**: We sent a copy of anonymised results to all participants along with their pseudonym and invited them to provide feedback on the accuracy of our thematic analysis and use of quotes.

## Results

### Participant characteristics

In total, we conducted 19 CE semi-structured interviews (6 men, 13 women) over ZOOM™ (N = 12), in-person (N = 5), and over the phone (N = 2). Interviews lasted 40 minutes on average. Participants' characteristics are described in Table 2. All CEs work full-time with individuals experiencing homelessness in Dublin and see high rates of co-occurring addiction (~80% of homeless service users). The five primary healthcare professionals interviewed represent four drop-in health and addiction care services available to PEH in Dublin inner city. Three CEs represent Dublin's hospital-led Inclusion Health services. Other interviews were held with government health agents (N = 2), researchers (N = 3), addiction services operations and nursing staff (N = 4), one social care professional (N = 1), and one psychotherapist providing crisis intervention to PEH (N = 1).

In the following sections, we focus on three main themes emerging from our analysis. The first two–'State of homelessness and addiction in Dublin' and 'Healthcare usage and outcomes'–provide context for the circumstances in which CEs are currently working. The final and most comprehensive theme–'Priority healthcare needs'–summarises CEs' perspectives on areas for improvement. For conciseness, we elected not to exemplify our analysis with data extracts but make references to the participants whose views underpin discursive claims throughout using their pseudonym (e.g., Homeless Health Services1, Hospital1).

### Theme 1 - State of homelessness & addiction in Dublin

CEs described the changing landscape of homelessness in the city. Though a ban on evictions reduced the number of people entering homelessness during the Covid-19 pandemic, housing statistics show a sharp increase as soon as the ban was lifted (Government1). The severity of the housing crisis has resulted in emerging homelessness unrelated to addiction. More migrants and asylum seekers, youth leaving social care, elderly individuals, women, and families are entering homelessness (Government1, Hospital1). For the first time in 2023, newly arrived asylum seekers are being discharged from hospital with no place to sleep due to a shortage of beds in homeless accommodation (Hospital3).

CEs noted that the nature of drug addiction is also changing. Today, most substance misuse in Ireland involves some combination of alcohol, cocaine, benzodiazepines, cannabis, and opiates. CEs primarily treat and respond to this type of polydrug use: *"a lot of people will probe back into heroin and [benzodiazepines] to come down from a crack cocaine binge. Very rarely I've met somebody who was going to the clinic and only taking methadone."* (Addiction

**Table 2. Community expert characteristics: Dublin, Ireland 2022–2023.**

| CE ID | Professional Role | Location | Interview Format |
|---|---|---|---|
| Homeless Health Services1 | Healthcare professional (Primary care) | Drop-in primary and addiction care | Zoom |
| Homeless Health Services2 | Healthcare professional (Primary care) | Drop-in primary and addiction care | Zoom |
| Homeless Health Services3 | Healthcare professional (Primary care) | Drop-in primary and addiction care | Zoom |
| Homeless Health Services4 | Healthcare professional (Primary care) | Drop-in primary and addiction care | In person |
| Homeless Health Services5 | Healthcare professional (Primary care) | Drop-in primary and addiction care | In person |
| Hospital1 | Healthcare professional (Hospital) | Hospital Inclusion Health service | Zoom |
| Hospital2 | Healthcare professional (Hospital) | Hospital Inclusion Health service | Zoom |
| Hospital3 | Healthcare professional (Hospital) | Hospital Inclusion Health service | Zoom |
| Government1 | Research and Data Officer | Health Service Executive Ireland | Zoom |
| Government2 | Project Manager | Health Service Executive Ireland | Zoom |
| Psychotherapist1 | Psychotherapist | Drop-in primary and addiction care | In person |
| Researcher1 | Researcher and Physiotherapist | Drop-in primary and addiction care | Zoom |
| Researcher2 | Researcher | Residential treatment facility | Zoom |
| Researcher3 | Researcher | Drop-in primary and addiction care Emergency department | Zoom |
| Social Care1 | Child and Family Support Network Coordinator | Government Child and Family Agency | Zoom |
| Addiction Services1 | Programme Coordinator | Drugs and Alcohol Task Force | Phone |
| Addiction Services2 | Team Leader | Dublin Harm Reduction NGO | In person |
| Addiction Services3 | Head of Service | Residential treatment facility | In person |
| Addiction Services4 | Clinical Nurse Specialist | Residential treatment facility | Phone |

Services1) Crack cocaine use has skyrocketed and is a particularly difficult problem as there are few effective treatment options (Homeless Health Services4, Addiction Services1). For other substances, Dublin's addiction care services offer a mixture of opiate substitution–mostly methadone with some suboxone prescribing–, alcohol detoxification, benzodiazepine detoxification, and more recently, benzodiazepine maintenance.

## Theme 2 - Healthcare usage & outcomes

Patterns of healthcare usage by PEH have been widely documented. CEs reiterated that when in homelessness, immediate health needs (e.g., addiction, housing, acute injury, infection) take precedence over long-term preventative needs. PEH have lower rates of medication adherence, tend to see the GP mostly for methadone, and to otherwise present to health services as a last resort. Internal barriers to presentation include mental health conditions, distrust of the health system, and internalised stigma, shame, and embarrassment. External barriers, particularly at the hospital level, include not getting methadone or Librium in a timely way, lack of freedom–particularly for those who have been in prison–, and judgement and blame from healthcare providers. Participant Hospital2 noted a tendency for transience across services. PEH may struggle to sustain a hospital admission due to their addiction or having co- diagnosis of a psychiatric disorder.

CEs working in drop-in primary care perform mostly acute medicine, often related to addiction. They see high rates of blood borne infectious diseases (i.e., HIV and Hep C), nutritional issues, psychiatric anxieties from taking street tablets, and physical–often injection-related–injury such as ulcers, cellulitis, or trauma. Comorbidity, physical frailty, and early ageing are common. Many patients have ischemic heart disease or chronic airways disease made worse by smoking heroin or crack cocaine along with comorbidity of injuries caused by injecting. Other presenting issues include musculoskeletal and dermatological conditions,

pregnancy, and high rates of depression and self-harm. CEs working in hospital settings see many patients in homelessness for deteriorating mental health conditions, overdose, orthopaedic surgical issues, or seizure disorder related to injury or substance use. CEs' patients experience high rates of childhood and adulthood trauma, and/or domestic violence that tend to precede problematic substance use.

## Theme 3 - Priority healthcare needs

Five key subthemes relating to priority healthcare needs were identified (Fig 1). Key quotes relating to each sub-theme are presented as S1 Table.

### 3.1 Promote culture that values health equity

To sustainably improve the provision of healthcare to PEH, CEs outlined a need to increase the social value attributed to health equity. Currently, health policies are designed and implemented in a cultural context that values tradition and evades risk to the detriment of marginalised populations. As a result, despite worsening health inequalities, long-time administrative barriers to healthcare for PEH persist (e.g., appointment systems, proof of address requirements, management of addiction in hospital) (Hospital1 and 3, Addiction Services4, Researcher3). CEs noted unanimously that the current standard of care excludes PEH and encouraged a shift towards proactive, innovative, patient-centred initiatives moving forward. This will require a cultural shift towards understanding that homelessness and addiction stem from systemic inequity rather than personal choice.

### 3.2 Accelerate action in addressing health inequalities

**Strategic planning, collaboration, and monitoring.** CEs noted an overreliance on reactive, shortsighted healthcare strategies implemented in response to co-occurring housing, addiction, and mental health crises. Instead, they hoped to see policymakers take the lead in developing a wider and more strategic approach to achieving equitable health. This will require the redistribution of resources in proportion to relevant needs and long-term funding allocation for innovative but evidence-based initiatives. Vulnerabilities specific to youth, women, migrants, and ethnic minority individuals in homelessness were a recurring theme throughout interviews, underlining the urgency of strengthening social protection policies for each group.

CEs reiterated that homeless individuals have long standing and sometimes chronic health issues which relate not just to housing but to a conglomeration of issues such as poverty, substance use, domestic violence, tragic accidents, and social isolation. Despite the intersection of issues, Researcher2 noted that historically in Ireland, substance use has been viewed and responded to separately from health. Today, according to CEs, homelessness, mental health, and addiction services continue to act as separate agencies with little communication between them. Many services (e.g., probation, housing, addiction, healthcare, mental health, child services) are involved in one person's care and communication is not always optimal. To address this, truly comprehensive case management and routine interagency meetings were recommended moving forward.

One aspect contributing to the compartmentalisation of care is a lack of data linkage between agencies. Separate electronic records systems and lack of individual patient identifiers impeded work, specifically. While homeless medical services use a shared system, that data is not accessible to normal GP practices and vice versa (Homeless Health Services4), nor are primary care records linked to secondary care records (Homeless Health Services5). In the hospital setting, lack of electronic patient identifiers contributes to repetition in care as repeat admissions are sometimes started again from scratch. Government1 outlined Health Service

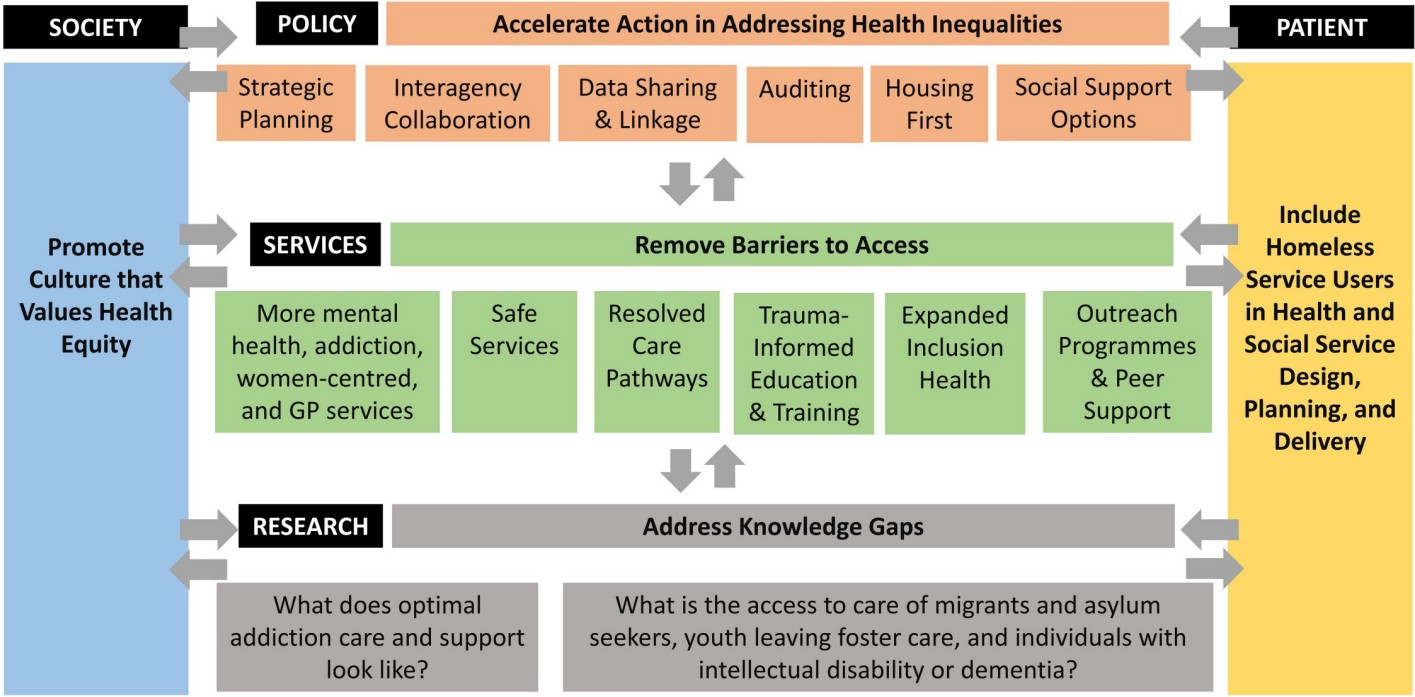

**Fig 1. Overview of healthcare priorities amongst individuals experiencing homelessness in Dublin: Community expert perspectives.**

Executive Ireland (HSE) Social Inclusion's plan to establish data sharing policies between different partner agencies and to look at linking homelessness administrative data with health data by 2026. CEs noted that accelerating these aims will be important to ensure that health policies are data driven in future.

Addiction Services1 and 3 commented on how a lack of objective quality control procedures result in a lack of standardisation and transparency. For example, the HSE and NGOs employ case managers to coordinate a service user's care plan across many services (e.g., housing, health, addiction, social services, justice). Yet, Addiction Services3 had noticed instances where purported case managers held a role similar to a key worker or project worker, only dealing with one aspect of a client's care. Addiction Services1 noted inconsistencies in reported progressions in housing (i.e., clients are just being moved from one hostel to another) and in drug assessments (i.e., a client's 'problem drug' is noted as heroin when in fact there is poly-drug use). In addition to impacting the quality of care, mislabelling can impact clients' convictions in court as their progress is assessed based on the health and housing supports received (Addiction Services3). Clients–who respond well to fairness and consistency after growing up in unpredictable environments–can further lose their already tenuous faith in the system (Homeless Health Services1).

**Housing first.** CEs emphasised that efforts to improve someone's health cannot work if that individual does not have safe and stable housing. Dissatisfaction with temporary accommodation for PEH in Dublin was mentioned by all CEs. Homeless hostels are often detrimental to health due to ubiquitous use of drugs, physical and sexual violence, noise, and lack of storage for medications (Addiction Services4, Hospital1 and 3). CEs mentioned clients who preferred prison or sleeping rough to temporary homeless accommodation (Homeless Health Services5, Addiction Services1). Some PEH–especially those without proof of Dublin residency–struggle to access temporary accommodation at all. Some PEH rely on the emergency

department (ED) as a safe place to sleep. Others are being discharged from hospital with no place to go.

Four CEs mentioned the effectiveness of the Housing First programme which, by providing permanent housing for people experiencing homelessness without any preconditions around addiction or mental health treatment, views housing as healthcare (Government1 and 2, Addiction Services1 and 4). CEs hoped to see an expansion of this programme (Government1 and 2, Addiction Services4) accompanied by more extensive wraparound supports that include daily living skills training (Addiction Services1).

**Social support options.** CEs felt that Ireland relies on an overmedicalized response to homelessness and its associated conditions that fails to address underlying causes such as loneliness, neglect, isolation, and grief. With limited avenues for social and emotional support, clients in homelessness often lack the self-esteem required for recovery.

CEs considered ways to reformulate how the system views, responds to, and prevents intersecting homelessness, addiction, mental health, and physical health impairments. Suggestions included having a societal conversation on the drivers of homelessness and addiction; focusing on the strengths rather than weaknesses unique to PEH including resilience, creative problem solving, honesty, humour, and altruism (Psychotherapist1, Hospital1); and–in more concrete terms–prioritising funding for community development initiatives. CEs hoped to see more spaces and opportunities for PEH to make social connections (e.g., adult education, exercise) (Psychotherapist1, Researcher1). For individuals for whom addiction impedes participation in social programmes, showing patience, compassion, and advocacy in the provision of healthcare was mentioned as a way to encourage self-confidence (Hospital1 and 3, Homeless Health Services1).

### 3.3 Remove barriers to access

**More and safer services.** Though health system flaws are not unique to the homeless health sector, service shortages disproportionately affect PEH who face worse health outcomes and increased barriers to care. Shortages of specialised homeless services in rural areas are contributing to system saturation in Dublin as clients come from all over the country to receive care. Lack of beds for psychiatry was highlighted as a "*huge, huge issue*" in Ireland (Hospital3, Addiction Services2), as was lack of accessible psychiatrists. Many patients struggle to receive mental health diagnoses. Others who are diagnosed but not at immediate risk to themselves or others may not be eligible for care. For those who don't fit criteria for admission, providers generally advise patients to "*go home to rest and be with family*" (Hospital3); advice that is inappropriate for someone in homelessness. Some GPs are responding to the emergency by prescribing psychotropic drugs without diagnosis.

Lack of drug treatment beds was of equal concern to CEs who see clients waiting between 3 and 6 months to enter a facility (Addiction Services1,2, and 4, Researcher3). Capacity constraints result in that most drug treatment facilities have clean urine requirements and maximum entry doses for methadone (Addiction Services1, 2 and 4). Patients in homelessness struggle to meet these requirements without additional support, yet stabilisation programmes that house people while they come down to make the criteria for other residential services are in short supply. Two stabilisation units now exist in North Dublin, but they are not enough to account for the explosion of crack cocaine use for which "*the most effective treatment option is to get somebody into a stabilisation residential admission.*" (Homeless Health Services4).

CEs wanted to see more women-centred services that respond to the intersection of substance use, domestic and sexual violence, homelessness, and motherhood. Specifically, there is a shortage of methadone and suboxone stabilisation programmes for pregnant women

(Addiction Services1 and 4). In terms of residential treatment, encouragingly Ireland is one of few countries across Europe with a facility for domestic violence victims active in their substance use, and two more for mothers with children under five (Researcher2). When capacity is reached, however, or for women with older children, recovery remains a challenge. Gender-specific spaces are also needed in homeless health services. Many homeless women choose to stay with a violent partner who offers them protection from other men and are likely to present to health services with that partner (Hospital1).

A recent shortage of GP availability is also impeding access to care (Addiction Services4, Homeless Health Services4, Hospital3). Homeless individuals who have an address are unable to step down from homeless health services to a normal GP practice with capacity to treat longer-term conditions. The system the HSE has in place requires that someone applies to and is refused from 3 GP practices before they are assisted in finding a provider. Hospital3 noted, "*for someone in homelessness, how are they going to do that unless somebody does it for them*?" As a result, CEs in Dublin are facing the dual burden of increased administrative work and navigating the mismatches between traditional referral/appointment systems and homelessness over the phone and via email (Hospital3). Facing overcrowding in their own clinics, homeless health providers require additional administration support in order to maximise time with patients (Hospital3). Currently, limited time necessitates that they focus primarily on acute conditions and addiction rather than chronic disease management.

CEs underlined how safety must be built into health services as well as temporary housing. In addition to safety issues connected to gender-based violence, the complexity of addressing safety issues related to drug debts and drug dealing in proximity to low threshold services was noted by CEs (Hospital1, Homeless Health Services4, Addiction Services1, Researcher3). To address safety issues beyond the control of the health system, CEs commented on the importance of having experienced operations staff as gatekeepers within homeless health services. The provision of training for healthcare staff to spot and report signs of intimidation or violence was also recommended (Addiction Services1). Homeless Health Services1 and Researcher3 also hoped to see conversations commence on how to create safe services for people trying to manage their addiction without creating additional barriers to care for people active in their drug use.

**Resolved, stigma-free care pathways.**   CEs highlighted that care pathways in response to 'crisis points' aren't fully resolved. Domestic and sexual violence, for example, often go undetected in hospital (Researcher2 and 3). The current system also fails to assess and support individuals who experience repeat overdose (OD) (Homeless Health Services3 and 5). CEs noted that identifying and supporting patients at these critical junctures could set them on a road to recovery, however this will require overcoming barriers to detection. At the patient level, fear of stigma and shame may prevent full disclosure of an event. At the provider level, underlying conditions and traumatic events may be put down to addiction and not explored further. At the system level, staffing shortages and lack of services result in that healthcare providers are expected to detect and respond to conditions with which they are unfamiliar. Overcrowding limits time with patients to build trust, detect a point of crisis, and refer to specialised care and support. Even if the referral stage has been reached, there may be no space available in the appropriate service.

Another unresolved care pathway relates to muti-morbidity. CEs found that patients with multiple conditions tend to get lost between health services. Though actions and pathways are improving around dual diagnosis of mental health and addiction issues (Researcher2, Addiction Services1 and 2), they're still not fully resolved (Researcher2 and 3, Addiction Services2) particularly when another layer of complexity is added (e.g., domestic violence, trauma). Parents whose conditions fall between the stools of other services can be reverted to child

protection (Social Care1, Addiction Services1); a pattern that has grave consequences for mothers who do not present to health services for fear of their child being taken into care (Hospital3, Researcher3). For mothers who *do* present, Researcher3 and Homeless Health Services3 spoke on the lack of designated support or pathway for homeless women facing addiction whose children are taken into care at birth.

A majority of CEs noted that widely cited attitudinal barriers to care persist in mainstream health services. Homeless patients regularly encounter disrespect, discrimination, blame, impatience, and judgement from healthcare providers (Hospital3, Researcher3, Addiction Services4). In Ireland, general healthcare professionals–despite the inevitability of encountering addiction in practice–receive very little education on its underlying causes. To address ongoing stigma, CEs recommended expanding trauma-informed care training in both university and medical settings, *"We're trying to discuss getting funding to do trauma-informed care for all staff, from cleaners to secretaries to porters to nurses to doctors. Across the board." (Hospital3)* The goal of these trainings would be for all professionals interacting with individuals experiencing homelessness and/or addiction to provide care imbedded in an understanding of trauma experiences and how they shape patients' views of and interactions with healthcare.

**Expanding patient-centred programmes.** CEs recommended expanding hospital Inclusion Health teams (Homeless Health Services 4 and 5, Hospital2 and 3). Two Dublin hospitals now offer dedicated services providing patient-centred, trauma-informed care to individuals experiencing homelessness. To prevent repeat presentations, the teams support patients to attend appointments, to create feasible discharge plans, and to access medications. At the primary-acute care interface, these services are vastly improving continuity of care by linking in with homeless primary health services about specific clients. The services are also set up to manage withdrawal in hospital, meaning patients are less likely to leave early and return to primary care with unresolved issues.

CEs also felt positive about outreach integrated care programmes like Ireland's Hepatitis C treatment programme which brought screening, diagnosis, and treatment out into the community. Managing chronic conditions through outreach programmes was of particular interest as they are difficult to treat in drop-in primary care or traditional hospital settings (Homeless Health Services4, Hospital2 and 3). Moving forward, CEs recommended testing the feasibility, acceptability, and eventual effectiveness of outreach programmes for epilepsy (St James's hospital have an outreach programme for epilepsy but it cannot match the quantity of homeless patients across Dublin with seizure disorder (Hospital2 and 3)), respiratory issues related to smoking crack cocaine (Hospital3, Homeless Health Services4), and diabetes (Hospital3). These programmes will have to anticipate and overcome challenges associated with long-term disease management within a transient population.

All CEs who mentioned potential outreach programmes noted the importance of involving peer support workers with lived experience in addiction and homelessness (Hospital2 and 3, Homeless Health Services1-3). CEs spoke of the potential value in expanding peer support roles to new programmes for both the employee and client. For the peer support worker, the job is an avenue for rebuilding confidence and routine on the other side of addiction. For the client, interacting with someone with shared experiences creates *"an identifiable sort of bond." (Homeless Health Services1).*

### 3.4 Address knowledge gaps

**Optimal addiction care and support.** Uncertainties linger surrounding appropriate addiction care and support in response to a changing drugs landscape. For example, CEs were divided in their views on methadone. Some viewed it as *"liquid handcuffs"* (Addiction

Services1 and 3), while others noted its effectiveness in preventing overdose and relapse (Homeless Health Services1). The question of suboxone versus methadone for the treatment of opioid dependence arose, as did the potential for Ireland to expand prescription of buprenorphine prolonged-release injection (Buvidal).

CEs held differing opinions on benzodiazepine prescribing. Addiction service providers referred back to their sense that GPs are overprescribing for addiction, whereas GPs had a sense that benzodiazepine intoxication has improved since the implementation of a maintenance programme during the Covid-19 pandemic. CEs did not entirely understand why, *"people seem to be dying to finish methadone, but don't want to finish benzodiazepines"* (Homeless Health Services1). Another 'unknown' relates to sleeping difficulties. GPs are unsure of how to respond to clients who strongly prefer to stay on sleeping tablets even after they are no longer effective (Homeless Health Services4).

Homeless Health Services1 also noted a potential gap in beliefs between doctors and those in addiction regarding recovery: *"Maybe we don't actually have the same aim in mind. Would patients seeking stability ultimately like to be drug-free or something else? Are we having honest conversations where patients feel they can share what their treatment goal would be?"*

**Understanding access amongst vulnerable populations.** CEs were careful to point out that they don't know about the health of individuals who do not interact with their services. Eight CEs mentioned migrants and asylum seekers in Ireland who are experiencing increasing rates of homelessness. Potential barriers to care amongst this population include language barriers and lack of rights to welfare, housing, and medical care. This group will also have specific health needs: "*a lot of them wouldn't have addiction issues or alcohol issues, but they'd have huge trauma from war*" (Hospital3). CEs were also concerned about the transition from adolescence to adulthood for teenagers leaving state care. Many are entering directly into homelessness at 18 (Hospital1, Government1), a situation that is challenging to navigate and extremely detrimental to health. With the ongoing housing crisis, some elderly people with dementia are going into homelessness after forgetting to pay their rent (Hospital1). People with intellectual disability are falling into homelessness because of their intellectual disability. Little is understood about each of these groups' trajectories, health needs, and how they access and interact with services.

Five CEs mentioned uncertainties relating to the health usage behaviour and access to care for Irish Travellers and Roma people in homelessness who face increased discrimination due to their ethnicity. Specific, intersecting identities may further shape health needs (Hospital1, Government1). Research gaps mentioned related to rough sleepers, lone parents in homelessness, homeless women engaging in sex work, and members of the LGBTQ+ community (Government1 and 2).

## 3.5 Include homeless service users in health and social service design, planning, and delivery

Policy and system-level flaws result in a lack of immediacy of care which *"is key when you're dealing with individuals that have a large degree of instability in their lives."* (Researcher2) In turn, many clients have lost faith in and disengaged from the system. Furthermore, to have available services and clear pathways in place does not imply that patients–especially those in great deals of emotional and/or physical pain–will have the trust or capacity to use them.

There was a general sense that resolving flaws in the system (e.g., availability of services, compartmentalised care, unresolved pathways) and reducing stigma through trauma-informed training will play a role in rebuilding trust and engagement. However, CEs noted a

simultaneous need to increase empowerment by providing PEH a participatory space through which to take part in decision-making about health services.

Traditionally, people in homelessness and addiction have little freedom of choice. Often as a result, patients do not feel in charge of their own health or a sense of entitlement to anything better (Hospital1). Providing service users with opportunities to have ownership and self-expression over their care can be a mechanism for re-instilling a sense of responsibility and engagement. Incorporating user voices will also improve the quality and effectiveness of services. CEs mentioned how well-versed people experiencing homelessness are in the system (Researcher3, Hospital3) and skills such as resourcefulness and creativity (Hospital1). Despite this, there is a lack of data looking at how PEH would like their own healthcare in Ireland (Hospital2).

Considerations moving forward include creating spaces for service users to provide feedback anonymously and without repercussion (Hospital1), creating opportunities for exchange between policymakers and service users (Addiction Services2, Social Care1), and accounting for potential literacy barriers (Hospital1) and the well-cited dilemma that, *"the individuals that need support the most, unfortunately, are the ones that may be least likely to engage."* (Social Care1)

## Discussion

This qualitative study was the first in Ireland to examine priority healthcare needs amongst PEH through consultation with community experts. Encouragingly, CEs' views are in line with internationally recognised community solutions for promoting health equity, including making health equity a shared vision and value, fostering multi-sector collaboration, and increasing community capacity to shape outcomes [28]. At the societal level, CEs identified a need to promote a culture that values health equity. In the United States, 'public will building' approaches have successfully integrated grassroots methods with traditional mass media tools to–by connecting to the existing, closely held values of target groups–encourage social change [29].

At the policy level, accelerating action in addressing health inequalities was recommended with an emphasis on strategic planning, Housing First, social support options, interagency collaboration, improved data linkage and sharing, and auditing. People with lived experience of homelessness and health professionals across Canada ranked access to housing and care coordination/case management as top priorities [13, 30, 31]. Findings from Toronto highlight a need for more services that encourage the integration of homeless individuals into social networks and the building of specific types of social support within networks [32]. Other policy needs are specific to the Irish context. For example, Ireland lags behind other countries in adopting digital health and remains one of three EU countries without a shared electronic health records system or individual health identifiers [33]. In terms of the potential effectiveness of CEs' recommendations, Housing First, with immediate provision of housing in independent units with support, improves outcomes for individuals with serious mental illnesses [34]. Standard case management with coordination of services improves mental health, housing, and addiction outcomes [34]. However, our findings re-iterate that the effectiveness of these interventions depends upon their availability and comprehensiveness; two conditions that CEs felt would be better upheld in response to routine, external auditing.

At Ireland's health services level, removing barriers to access will require the provision of more mental health, addiction, women-centred, and GP services; the embedding of safety within those services; resolved care pathways in relation to crisis points and multi-morbidity; trauma-informed education and training for university medical students, clinical and non-clinical staff; expanded hospital-led Inclusion Health programmes; and, potentially, outreach

programmes and peer support for chronic disease management. Evidence shows that primary health-care programmes specifically tailored to homeless individuals might be more effective than standard care and are more likely to achieve higher patient-rated quality of care [34]. Ireland, like neighbouring Scotland, follows this model by providing specialist homeless health services. However, the success of this approach requires that PEH are able to move back into mainstream services when their crisis is resolved [35]. CEs described a scenario in Ireland where individuals who have experienced homelessness and/or addiction cannot re-integrate into mainstream services due to shortages of GPs and mental health professionals. Resolving these shortages will require addressing disproportionately high rates of emigration among Irish medical graduates and junior doctors [36].

In other areas, Ireland is leading innovation at the community level. The 'HepCheck Dublin' programme mentioned by CEs provides a model for a community-based treatment approach [37]. The North Dublin City General Practitioner Training Programme educates doctors to work with marginalised, urban populations and has increased empathy and decreased prejudice amongst trainees [38]. Very few hospitals internationally offer dedicated Inclusion Health services like the two in Ireland that, by providing a more coordinated approach to patient care, are improving patient outcomes [39]. Incorporating this type of patient-centred programmes into strategic healthcare planning could be a way to remove barriers to care in Ireland and abroad. Similar to our findings, community experts and members from Canada, the US, and the UK identified a need for better chronic disease management, expanded clinical outreach programmes, trauma-informed training, and improved referral pathways for PEH [12, 13, 40].

CEs highlighted knowledge gaps relating to optimal addiction care and support and access to healthcare amongst vulnerable populations that are not unique to Ireland. There is international debate regarding the desirability of interventions that emphasise abstinence versus harm reduction approaches for substance misuse [34]. Evidence suggests homelessness is a risk for and consequence of Alzheimer's disease and related dementia (ADRD) but there is a dearth of studies on the intersection of homelessness and ADRD [41]. There is also an identified a need to improve the knowledge base around the health and housing needs of individuals and children with intellectual disability (ID) and/or autism [42]. To our knowledge, this is the first study to document the need to assess health needs amongst homeless asylum seekers and other migrants in Ireland. Incorporating service users' perspectives will be vital for answering these research questions. However, our study findings emphasise that the voices of PEH, including representatives from specific homeless groups, must also be embedded more deeply into health and social service design, planning, and delivery to facilitate lasting change.

## Limitations

As this study was located in one city, findings may not be transferrable to other locations with different health systems. That being said, we identified crossover with priority healthcare needs identified amongst PEH in Canada, the US, and the UK, indicating the potential relevance of findings to other countries with similar specialist service approaches to homeless healthcare. Our choice of a qualitative approach to community health needs assessment means that we will not have identified all healthcare needs of PEH in Dublin, instead focusing on timely insights. Readers are encouraged to use this document as a starting place for understanding priority system flaws and unmet needs rather than an exhaustive list. Furthermore, this study reports the views of service providers which may differ from those of service users. Our research team intends to report homeless service users' views for comparison in a subsequent study. Finally,

though no new themes emerged during the last few interviews and we felt that our data had reached saturation, it is possible that a larger sample may have yielded more themes.

## Conclusion

Qualitative interviews conducted with community experts revealed five key healthcare priorities for homeless individuals in Dublin at the societal, policy, health services, and patient levels. In the short-term, findings reveal an urgent need for more Housing First tenancies, inpatient stabilisation and residential treatment beds, acute mental health beds, and women-centred services. Essential posts in mental health and mainstream general practice services must be filled to ensure that individuals who are no longer in crisis can step down from specialised homeless to mainstream health services. In the long-term, CEs considered ways to reformulate how the system responds to intersecting homelessness, addiction, mental health, and physical health impairments including prioritising funding for community development initiatives and social support. The success of future reforms will require establishing a national culture that values health equity and incorporating the voices of people experiencing homelessness at all levels of service design and delivery. Priority research needs include understanding service users' views on optimal addiction care and support and investigating access to healthcare for homeless asylum seekers and migrants, youth leaving foster care, and individuals with intellectual disability or dementia.

## Supporting information

**S1 Table. Supporting quotes for priority healthcare needs themes and sub-themes.**
(DOCX)

**S1 Text. Data supporting study results.** Anonymised excerpts from semi-structured interview transcripts.
(PDF)

**S2 Text.**
(PDF)

## Author Contributions

**Conceptualization:** Carolyn Ingram, Conor Buggy, Carla Perrotta.

**Data curation:** Carolyn Ingram.

**Formal analysis:** Carolyn Ingram, Isobel MacNamara, Conor Buggy, Carla Perrotta.

**Funding acquisition:** Carla Perrotta.

**Investigation:** Carolyn Ingram.

**Methodology:** Carolyn Ingram, Carla Perrotta.

**Project administration:** Carolyn Ingram.

**Resources:** Carolyn Ingram.

**Supervision:** Conor Buggy, Carla Perrotta.

**Validation:** Carolyn Ingram, Conor Buggy, Carla Perrotta.

**Visualization:** Carolyn Ingram.

**Writing – original draft:** Carolyn Ingram.

**Writing – review & editing:** Carolyn Ingram, Isobel MacNamara, Conor Buggy, Carla Perrotta.

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
