## [Decision Letter · Decision Letter 0]

11 Sep 2023

PONE-D-23-25256Priority healthcare needs amongst people experiencing homelessness in Dublin, Ireland: a qualitative evaluation of community expert experiences and opinionsPLOS ONE

Dear Dr. Ingram,

Thank you for submitting your manuscript to PLOS ONE. After careful consideration, we feel that it has merit but does not fully meet PLOS ONE’s publication criteria as it currently stands. Therefore, we invite you to submit a revised version of the manuscript that addresses the points raised during the review process.

We look forward to receiving your revised manuscript.

Kind regards,

Adetayo Olorunlana, Ph.D.

Academic Editor

PLOS ONE

Journal Requirements:

4. Please ensure that you include a title page within your main document. You should list all authors and all affiliations as per our author instructions and clearly indicate the corresponding author.

5. We note that [Figure 1] in your submission contain [map/satellite] images which may be copyrighted. All PLOS content is published under the Creative Commons Attribution License (CC BY 4.0), which means that the manuscript, images, and Supporting Information files will be freely available online, and any third party is permitted to access, download, copy, distribute, and use these materials in any way, even commercially, with proper attribution. For these reasons, we cannot publish previously copyrighted maps or satellite images created using proprietary data, such as Google software (Google Maps, Street View, and Earth). For more information, see our copyright guidelines: http://journals.plos.org/plosone/s/licenses-and-copyright.

Reviewers' comments:

Reviewer's Responses to Questions

**Comments to the Author**

1. Is the manuscript technically sound, and do the data support the conclusions?

Reviewer #1: Partly

Reviewer #2: Yes

2. Has the statistical analysis been performed appropriately and rigorously? 

Reviewer #1: N/A

Reviewer #2: N/A

3. Have the authors made all data underlying the findings in their manuscript fully available?

Reviewer #1: Yes

Reviewer #2: Yes

4. Is the manuscript presented in an intelligible fashion and written in standard English?

Reviewer #1: Yes

Reviewer #2: Yes

5. Review Comments to the Author

Reviewer #1: Thank you for this opportunity to read and review this paper, which reports on an analysis of qualitative interviews conducted with experts working with homeless people in Dublin.

The paper is well set-up with a clear and informative account of the public health challenge of homelessness globally, and in Ireland particularly. Design and methods are laid out in transparent fashion, including with reference to the larger ethnographic study from which this paper arises. I note that a future paper, drawing on observational data, will follow in due course and I can see how the two manuscripts will sit will together. I am also entirely persuaded of the value of qualitative research in studies of this type, and I appreciate the attention to the approach taken to the management and analysis of data. I also like how information is included on the length of interviews, and the characteristics of the 19 interviewees.

However:

For me, the paper is unusually structured, and in this regard there is room for revision. As an aside, there needs to be some relabelling of figures, as Fig 2 is used in the text to refer both to the map of services and to the summary of themes. The reporting of findings begins in a conventional way on page 6 (using the authors' running footer manuscript page numbers), with two thematic headings ('State of homelessness...' and 'Healthcare usage...'). The next thematic heading, 'Priority healthcare needs', appears with an underlining which for me implies it has a thematically different status compared with the first two themes. This is not entirely clear, though. The 'Priority...' themes and sub-themes, with sample data extracts, are then displayed in a table, with recommendations alongside. This strikes me as an unusual way of presenting what might more usually be a rich, thematic, qualitative analysis. I also think that, at this point in the paper, it is far too early to be adding recommendations. I am also then somewhat thrown by the textual summary of these same tabulated themes, without data extracts, from page 15 onwards.

Overall, then, I think a more conventional presentation of thematic findings followed by a considered set of recommendations arising from these would work better and would also make for a more succinct read without the need to cross-refer the content of Table 3 and the textual summary beneath. An option here would be, I think, to avoid the use of Table 3 altogether.

I hope these comments prove useful.

Reviewer #2: Priority healthcare needs amongst people experiencing homelessness in Dublin,

Ireland: a qualitative evaluation of community expert experiences and opinions

This well written article describes a qualitative approach to needs assessment for the unhoused population in Dublin, Ireland. The article provides a strong rationale for the study, description of methods, analysis and incorporation of the literature into the discussion. I suggest the following to improve

1. Background: Since the article is centered on the concept of equity, I would appreciate a better understanding of equity issues in the drivers of the unhoused or in the provision of services that need to be addressed. For example, the authors state that substance use is the main cause of homelessness, but do not mention the structural and societal factors that contribute to substance use. Better defining what equity means in this context would be helpful.

2. Methods: While I agree that qualitative approaches are essential for understanding the needs of marginalised persons, I would recommend reframing the use of qualitative methods as something novel in needs assessments. Rather, most needs assessments are considered inadequate without the incorporation of community voices and perspectives on needs through qualitative engagement. There is considerable literature on how to do needs assessments that makes this clear. It would be interesting, for example, to include a review of the literature on how perspectives of the unhoused are incorporated into needs assessments, including reaching out to service providers.

3. Data Collection: The authors refer to the ethnographic study of the unhoused as part of the larger study and how this influenced the development of questions. This is very interesting, but if it is going to be included more information on how these two activities informed each other is important. Did the interview questions evolve over time as researchers gained more perspective on housing issues? Additionally, there is some confusion about the identification of interviewees since the data collection section describes being informed by the observations, while the recruitment refers to using a directory of service providers.

4. Results: The results reflect a comprehensive analysis and thematic coding framework. However, Table 3 needs to be revised. The length is a bit overwhelming. More importantly, the quotations included for some of the subthemes don’t really describe the name of the subthemes. The first sub theme for example, includes a quote that describes the need for a change in mindset but doesn’t mention health equity, thus it appears that the coders are making an interpretive leap. I would say the same for the theme strategic planning. It seems that the code is named more for the solution rather than the description of the issue. Since this is the results table, it is important to learn the perspectives of the service providers and where the commonality across different types of providers, perhaps. In a related issue, the recommendation column and references to it belong in the discussion section. The recommendations are a response to the data and reflect the authors’ interpretation of practice and policy approaches to improve health equity for the unhoused. I would include a separate table, or perhaps just discuss recommendations in the discussion section. Since the paper is addressing very complex issues, this would also demonstrate how solutions work together to solve problems, rather than addressing one issue at a time.

5. The discussion section is very informative and relates well to the literature on the subject. I would recommend a review of the article to ensure active voice in the methods and analysis description. The authors do a nice job describing each person’s role, but I always appreciate understanding who the authors are in relationship to the population and the study.

6. PLOS authors have the option to publish the peer review history of their article (what does this mean?). If published, this will include your full peer review and any attached files.

Reviewer #1: No

Reviewer #2: No

---

## [Author Response · Author response to Decision Letter 0]

28 Sep 2023

We have responded to each of the reviewers' comments in a separate file 'Response to Reviewer Comments'.

---

## [Decision Letter · Decision Letter 1]

9 Oct 2023

PONE-D-23-25256R1Priority healthcare needs amongst people experiencing homelessness in Dublin, Ireland: a qualitative evaluation of community expert experiences and opinionsPLOS ONE

Dear Dr. Ingram,

Thank you for submitting your manuscript to PLOS ONE. After careful consideration, we feel that it has merit but does not fully meet PLOS ONE’s publication criteria as it currently stands. Therefore, we invite you to submit a revised version of the manuscript that addresses the points raised during the review process.

We look forward to receiving your revised manuscript.

Kind regards,

Adetayo Olorunlana, Ph.D.

Academic Editor

PLOS ONE

Journal Requirements:

Reviewers' comments:

Reviewer's Responses to Questions

**Comments to the Author**

1. If the authors have adequately addressed your comments raised in a previous round of review and you feel that this manuscript is now acceptable for publication, you may indicate that here to bypass the “Comments to the Author” section, enter your conflict of interest statement in the “Confidential to Editor” section, and submit your "Accept" recommendation.

Reviewer #1: (No Response)

Reviewer #2: All comments have been addressed

2. Is the manuscript technically sound, and do the data support the conclusions?

Reviewer #1: Yes

Reviewer #2: Yes

3. Has the statistical analysis been performed appropriately and rigorously? 

Reviewer #1: N/A

Reviewer #2: N/A

4. Have the authors made all data underlying the findings in their manuscript fully available?

Reviewer #1: Yes

Reviewer #2: Yes

5. Is the manuscript presented in an intelligible fashion and written in standard English?

Reviewer #1: Yes

Reviewer #2: Yes

6. Review Comments to the Author

Reviewer #1: Thank you for taking the time to respond to the reviewer comments, and for summarising the changes you have made to your manuscript. My main observation related to the use of a table to summarise parts of your findings, and the overlap between the information here and the sections following in the original manuscript. For me, the paper is more clearly structured now, with main findings presented (by choice, without the use of extensive supporting data extracts) in the body of the article and with recommendations subsumed within the discussion section.

My final thought is, though: would it be an idea to introduce the thematic structure after table 2 (which helpfully summarises the participants), as opposed to heading directly into the section titled, 'State of homelessness...'? This would serve to introduce the overarching framework under which the analysis is progressed. It might also help to explain that you have elected to not exemplify your analysis with data extracts, but make references along the way to the participants whose data underpin the discursive claims you make. At present, although the meaning is inferred there is no statement explaining what you mean by including phrases like 'Hospital3' in your findings.

I hope these final observations are helpful.

Reviewer #2: The authors revisions have substantially improved the manuscript and I appreciate the enhanced definition of equity in conducting the assessment of the unhoused, as well as the removal of the recommendations in the results section. I agree that the discussion is comprehensive and well written. I am still confused by the headings "state of homelessness and addiction in Dublin" and "health care usage and outcomes" as distinct from "priority health care needs" with a thematic analysis, I would expect all three to be a theme, or not to have a heading listed that way. Please decide present these three topics as themes discussed in the interviews. If health care priorities has subthemes and the others don't that is fine. I think breaking it down into sub sub themes is excessive, as mentioned previously, I would use the heading under the sub theme 2 "accelerate action in addressing health inequalities" and describe the different points that emerged under that heading and take out all the subthemes. I would remove the parentheses (policy) and (level: health services). They are not included under every theme and they are not adding much and make the results too busy when I really just want to read about what people said in the interviews. My overall comment from the first review is not to over parse the data. The summaries are very descriptive, they fit well under the above theme, and do not need to be further broken down for the reader I think it is enough that the authors discussed the use of the framework in the analysis and if they want to return to it in the discussion that is appropriate. The size of the font in the headings also seems inconsistent with journal style, but I am sure that will be finalized in the editing process.

7. PLOS authors have the option to publish the peer review history of their article (what does this mean?). If published, this will include your full peer review and any attached files.

Reviewer #1: No

Reviewer #2: No

---

## [Author Response · Author response to Decision Letter 1]

25 Oct 2023

Thank you for your feedback. We've responded to each comment in the 'Response to Reviewers' document within the 'Attach Files' section.

---

## [Decision Letter · Decision Letter 2]

27 Oct 2023

Priority healthcare needs amongst people experiencing homelessness in Dublin, Ireland: a qualitative evaluation of community expert experiences and opinions

PONE-D-23-25256R2

Dear Dr. Ingram,

We’re pleased to inform you that your manuscript has been judged scientifically suitable for publication and will be formally accepted for publication once it meets all outstanding technical requirements.

Kind regards,

Adetayo Olorunlana, Ph.D.

Academic Editor

PLOS ONE

Additional Editor Comments (optional):

Reviewers' comments:

Reviewer's Responses to Questions

**Comments to the Author**

1. If the authors have adequately addressed your comments raised in a previous round of review and you feel that this manuscript is now acceptable for publication, you may indicate that here to bypass the “Comments to the Author” section, enter your conflict of interest statement in the “Confidential to Editor” section, and submit your "Accept" recommendation.

Reviewer #1: All comments have been addressed

Reviewer #2: All comments have been addressed

2. Is the manuscript technically sound, and do the data support the conclusions?

Reviewer #1: Yes

Reviewer #2: Yes

3. Has the statistical analysis been performed appropriately and rigorously? 

Reviewer #1: N/A

Reviewer #2: N/A

4. Have the authors made all data underlying the findings in their manuscript fully available?

Reviewer #1: Yes

Reviewer #2: Yes

5. Is the manuscript presented in an intelligible fashion and written in standard English?

Reviewer #1: Yes

Reviewer #2: Yes

6. Review Comments to the Author

Reviewer #1: Thank you for this final round of revisions, which relate to the introduction to the thematic presentation of findings.

Reviewer #2: I really appreciate the extensive background on the structural antecedents of homelessness and the changes to the presentation of the data. This is now a much more powerful paper.

7. PLOS authors have the option to publish the peer review history of their article (what does this mean?). If published, this will include your full peer review and any attached files.

Reviewer #1: No

Reviewer #2: **Yes: **Maia Ingram

---

## [Editor Report · Acceptance letter]

5 Dec 2023

PONE-D-23-25256R2 

Priority healthcare needs amongst people experiencing homelessness in Dublin, Ireland: a qualitative evaluation of community expert experiences and opinions 

Dear Dr. Ingram:

I'm pleased to inform you that your manuscript has been deemed suitable for publication in PLOS ONE. Congratulations! Your manuscript is now with our production department. 

Kind regards, 

on behalf of

Associate Professor Adetayo Olorunlana 

Academic Editor

PLOS ONE